

# Effects of soy isoflavones on menopausal symptoms in perimenopausal women: a systematic review and meta-analysis

Haodi Luan[1], Qianqian Liu[2], Yahui Guo[1], Hua Fan[1], Sileng A.[3] and Jing Lin[1]

[1] Clinical Nutrition Department, Critical Care Medicine Department, The Affiliated Hospital of Inner Mongolia Medical University, Hohhot, China
[2] Department of Gastroenterology, The Affiliated Hospital of Inner Mongolia Medical University, Hohhot, China
[3] Medical Simulation Center, Inner Mongolia Medical University, Hohhot, China

## ABSTRACT

Soy isoflavones are phytoestrogens found mainly in soy and its derivatives. Given their estrogen-like and antioxidant-inhibiting inflammatory effects, they have been hypothesized to be effective in treating menopausal symptoms. We conducted a systematic review in accordance with the PRISMA guidelines. In October 2024, we screened 2,099 articles, of which 12 were eligible for meta-analysis, and found that soy isoflavones were effective for treating menopausal symptoms (seven studies, 533 participants, Hedges' g $= -0.25$, 95% CI [$-0.42$ to $-0.08$], $p = 0.00$). Soy isoflavones showed significant effects on headache (three studies, 340 participants, Hedges' g $= -0.38$, 95% CI [$-0.60$ to $-0.17$], $p = 0.00$), psychosocial symptoms (five studies, 416 participants, Hedges' g $= -0.29$, 95% CI [$-0.48$ to $-0.10$], $p = 0.00$), palpitation (three studies, 356 participants, Hedges' g $= -0.42$, 95% CI [$-0.63$ to $-0.22$], $p = 0.00$), and depression (four studies, 748 participants, Hedges' g $= -0.72$, 95% CI [$-1.17$ to $-0.28$], $p = 0.00$), but no significant treatment effect on paresthesia symptoms, fatigue symptoms, physical symptoms, hot flushes, excessive sweating, insomnia, and vasomotor symptoms was observed. However, our results should be interpreted with caution owing to the small sample size. More trials should be conducted in the future to validate our findings.

## INTRODUCTION

Menopause is a natural biological process, marking the cessation of a woman's reproductive ability. It is characterized by various physiological and psychological changes. A very common and distressing symptom of this transitional period is the occurrence of vasomotor symptoms such as hot flashes and night sweats (*Avis et al., 2001*). The reduction in estrogen levels during menopause can lead to various symptoms that have a substantial impact on a woman's overall quality of life (*Groeneveld et al., 1993*). At present, hormone replacement therapy (HRT) is frequently used to alleviate menopausal symptoms such as hot flashes, night sweats, and mood fluctuations (*Pop et al., 2023*). However, despite its effectiveness, HRT is associated with several drawbacks, which include increased risk of cardiovascular

Corresponding author
Jing Lin, LinJing116@outlook.com

disease (*Cagnacci & Venier, 2019*), breast cancer (*Stoer et al., 2024*), and thromboembolic events (*Gomes & Deitcher, 2004*). Many women show concerns regarding the long-term effects of HRT, resulting in a preference toward alternative therapies. Consequently, there is growing interest in exploring non-hormonal treatment options, such as the use of plant-based isoflavones, particularly those derived from soy, which may provide symptom relief without the associated risks of traditional HRT. Overall, the transition toward safer, more sustainable treatment options reflects a significant shift in menopausal symptom management among perimenopausal women (*Kang et al., 2022*; *Nahas et al., 2007*; *Sohn et al., 2021*).

Soy isoflavones, a group of phytoestrogens predominantly found in soybean and its derivatives, have emerged as a potential alternative therapy for managing menopausal symptoms. These compounds, which primarily include genistein, daidzein, and glycitein, share structural similarities with human estrogen (17β-estradiol) and can bind to estrogen receptors, particularly ER-β. These compounds can therefore exert weak estrogenic effects (*Poschner et al., 2017*). When estrogen levels decline in menopausal women, soy isoflavones bind to estrogen receptor sites in the body. This binding induces a mild estrogenic effect, helping compensate for hormonal deficiencies and alleviate common menopausal symptoms such as hot flashes, night sweats, and mood swings (*Khapre, Deshmukh & Jain, 2022*). Conversely, during periods of high estrogen levels, soy isoflavones competitively block these receptors, reducing the effects of excess estrogen (*Viscardi et al., 2025*). This dual modulatory mechanism helps regulate hormonal balance and reduces discomfort caused by hormonal fluctuations during menopause. Although this mechanism suggests promising therapeutic potential, the clinical efficacy of soy isoflavones remains controversial. Some studies report substantial improvements in vasomotor symptoms and other menopausal disorders, whereas others show negligible or no benefits (*Chen, Ko & Chen, 2019*; *Chen, Lin & Liu, 2015*; *Gencturk, Bilgic & Kaban, 2024*; *Mainini et al., 2024*). These inconsistencies may be attributed to various factors, including differences in isoflavone formulations, dosages, individual metabolism (particularly the ability to produce equol from daidzein), and genetic variations in the study population. In addition, concerns have been raised about potential risks, such as interactions with endocrine function and breast cancer risk, although current evidence suggests safety at typical dietary intake levels (*Chen, Ko & Chen, 2019*; *Chen, Lin & Liu, 2015*; *Gencturk, Bilgic & Kaban, 2024*; *Mainini et al., 2024*). The conflicting results from numerous clinical trials and meta-analyses have sparked ongoing debates about the role of soy isoflavones in managing menopausal symptoms among the scientific community. Existing meta-analyses primarily investigate the association between soy isoflavones and menopausal symptoms as a whole, without addressing their effects on individual symptoms (such as insomnia, depression, hot flashes, and fatigue).

We aim to elucidate the role of soy isoflavones in managing menopausal symptoms, providing evidence-based guidance for women experiencing perimenopause, and fostering informed choices about dietary and supplementary options. By conducting a thorough systematic review and meta-analysis, we intend to deliver definitive insights into the effectiveness of soy isoflavones, by assessing the potential of these natural compounds to complement holistic strategies for women's health during menopause.

## METHODS

### Literature search strategy

We followed the PRISMA Statement guidelines to perform a systematic search (*Moher et al., 2009*). This study is also registered on the PROSPERO platform (CRD42024616691). The following databases were searched from inception to October 20, 2024: PubMed, Cochrane, Web of Science, and Embase. The following combinations of keywords were used during the search: soy, soya, flavones, flavonoids, genistein, daidzein, red clover, menopausal symptoms, and climacteric. Further, we hand-searched the reference lists of all included systematic reviews and meta-analyses to identify additional articles.

### Selection procedure

Studies were included if they met the following inclusion criteria: (1) a randomized controlled trial (RCT) design, including a parallel or crossover design; (2) inclusion of perimenopausal or postmenopausal women aged ≥35 years and experiencing menopausal symptoms; (3) inclusion of intervention group containing soy isoflavone (*e.g.*, soy extract, genistein, or isoflavone) and a control group with a placebo; (4) oral administration during the intervention; and (5) English language. The following studies were excluded: (1) observational studies, non-clinical trials, or animal studies; (2) studies where numerical outcome data were not provided; (3) studies that did not report on the specified outcome measures of interest.

### Data extraction

Two reviewers (Liu and Guo) independently and concurrently extracted data. Any uncertainties were resolved through consultation with a third reviewer (Luan). A standardized data extraction form was employed to gather information from the selected studies. We collected details regarding study characteristics, sample demographics, the type and duration of interventions, and the outcome measures. In addition, we obtained pre- and post-treatment means, standard deviations, participant counts, or pre-post mean differences and standard deviations for the purpose of meta-analysis. We also reached out to study authors *via* email to seek any missing data or clarifications, providing them with a tailored data table to facilitate the reporting of the requested information.

### Outcome measures

Outcome measures included menopause symptom-specific index scores (*e.g.*, Kupperman Index (*Kupperman, Wetchler & Blatt, 1959*), Greene Menopause Scale (*Greene, 1998*)), related menopausal symptoms, including hot flashes, insomnia, excessive sweating, headache, and fatigue, and quality of life scores (*e.g.*, MENQOL (*Hilditch et al., 1996*) and SF-36 (*Larson, 1997*)). An elevated score corresponded to a greater intensity and frequency of symptoms.

### Statistical analysis

We employed a random-effects model to compute the effect size. For continuous outcomes, we aggregated Hedges' g for adjusting the effect size for small sample sizes. We calculated 95% confidence intervals for each effect size estimate. The assessment of heterogeneity

among studies was conducted using the Cochrane Q and I² statistics. The thresholds for interpreting heterogeneity aligned with those established by the Cochrane Collaboration (I² 0–40%, possibly unimportant; 30–60%, potentially moderate heterogeneity; 50–90%, likely substantial heterogeneity; and 75–100%, significant heterogeneity) (*Higgins & Cochrane Collaboration, 2019*).

When multiple scales were used to measure the same symptom cluster, we selected the group that demonstrates superior data. When there were multiple treatment groups in a study (*Costa et al., 2017*; *Fontvieille, Dionne & Riesco, 2017*), we chose the group with soy isoflavone intervention plus exercise activities as the active treatment group, and the group with placebo alone plus exercise activities as the control group. One study (*Jou et al., 2008*) categorized participants into two groups based on whether their gut bacteria can break down and utilize soy isoflavones, producing the metabolite equol. In that study, we selected the data from the group that can metabolize and produce equol. For the assessment of menopausal symptoms, including hot flashes, hyperhidrosis, and insomnia, we used scoring systems rather than frequency counts. In these metrics, elevated scores corresponded to increased symptom severity.

We also examined publication bias *via* funnel plot analysis and used Egger's regression test to quantify the potential asymmetry of the plot. To address the issue of missing data, we used multiple imputation techniques to estimate missing values, ensuring the robustness of our findings. Subgroup analyses were performed based on participant characteristics and baseline disease severity to evaluate differential treatment impacts across diverse groups. Sensitivity analyses were conducted to determine the robustness of the results by repeating the analyses after removing studies with a high risk of bias. Meta-regression analyses were also employed to identify potential sources of heterogeneity, such as study design, sample size, and duration of intervention. All statistical tests were two-sided and a *p*-value < 0.05 was considered statistically significant.

## RESULTS

### Search results

Our literature search identified 2,099 publications, the specific search strategy can be found in the attachment. After removing duplicates, 1,204 titles and abstracts were screened. Further, full-texts of 41 articles were read for a detailed evaluation. Finally, 12 articles were included in this systematic review and meta-analysis. Reasons for exclusion have been explained in the PRISMA flow-chart (Fig. 1).

### Characteristics of included studies

We systematically evaluated the 12 studies (10 double-blind RCTs and two open-label studies (*Jou et al., 2008*; *Tranche et al., 2016*)) that investigated the effects of soy isoflavones and related compounds on menopausal symptoms, with a focus on climacteric symptoms such as hot flashes and depressive symptoms. These studies encompassed various research designs, including randomized controlled trials and observational studies. The sample sizes in these studies ranged from 32 to 192 participants, involving over 1,000 participants in total. The geographical distribution was extensive, covering North America, South

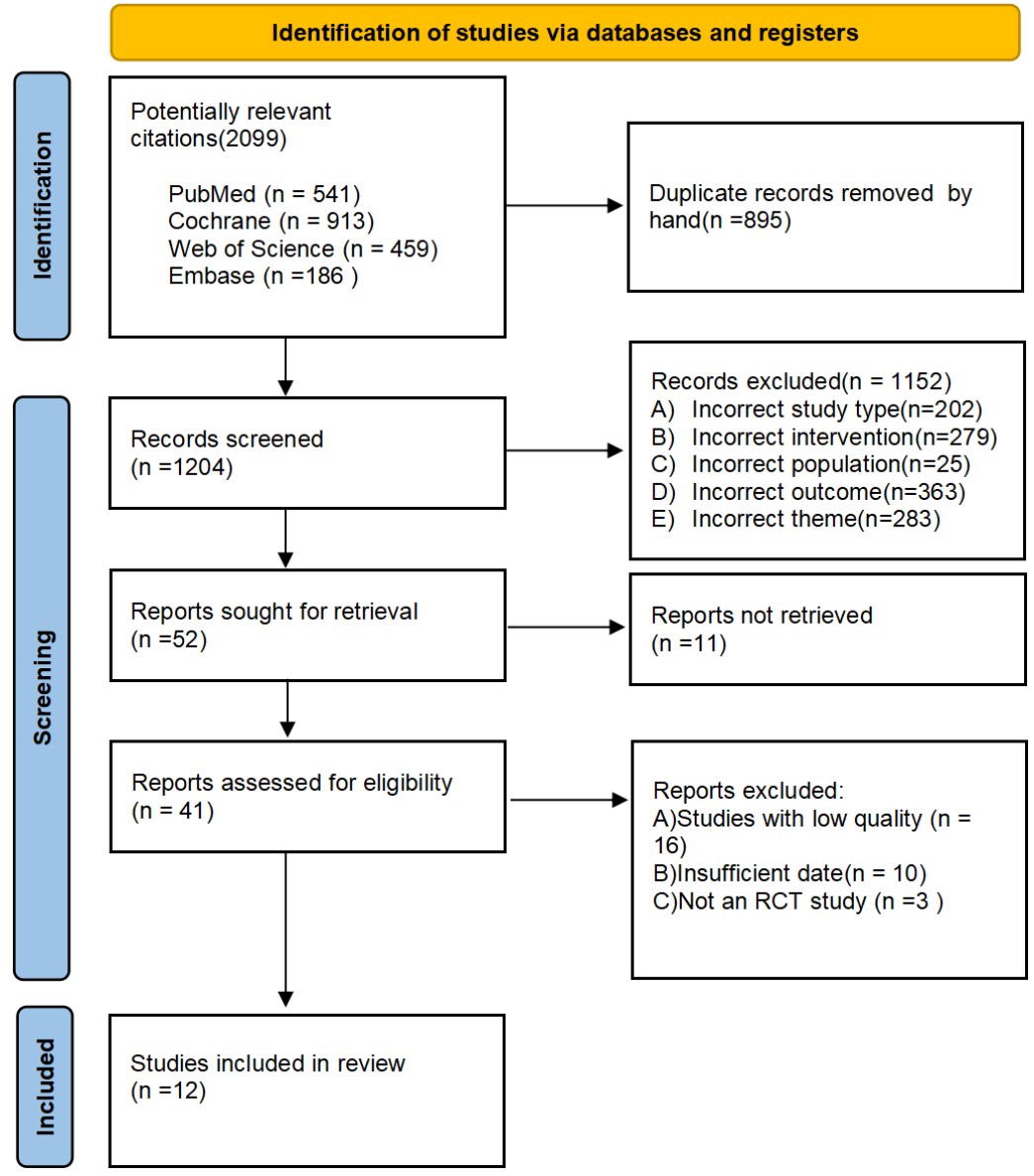

**Figure 1** **Prisma flow diagram of study selection process.**

America, Asia, and Europe, with five studies conducted in Asia, which may reflect the application and effects of soy foods in different cultural contexts. All studies focused on postmenopausal women. Herein, seven studies specifically evaluated the impact of soy isoflavones on menopausal symptoms, whereas the other five studies explored the comprehensive effects of soy extracts on the quality of life, bone health, and cardiovascular health of postmenopausal women. Interventions included soy protein, soy isoflavone supplements, and soy beverages, with daily intervention doses ranging between 30 mg and 209 mg of soy isoflavones. The control group was usually administered a placebo. The intervention duration varied between 12 weeks and 2 years, providing a basis for

assessing long-term effects. The primary outcomes included health-related quality of life, measured using instruments such as the Short Form-36 (SF-36) and MENQOL, climacteric symptoms assessed *via* the Kupperman Index and Greene Climacteric Scale, and depression symptoms assessed using the Zung Self-rating Depression Scale (ZSDS) and CES-D. The characteristics of the included studies are reported in Table 1.

## Risk of bias assessment

The included studies were assessed for risk of bias using the Cochrane Risk of Bias Tool by two independent reviewers, Fan and Lin, with discrepancies resolved through discussion with Guo. In random sequence generation, the studies by *Jou et al. (2008)* and *Tranche et al. (2016)* were rated as high risk owing to their open-label design. In allocation concealment, studies by *Jou et al. (2008)* and *de Sousa-Munoz & Filizola (2009)* were rated as unclear owing to lack of allocation details. In blinding of participants and personnel, studies by *Frigo et al. (2022)* and *Tranche et al. (2016)* were rated high risk because researchers knew the group assignments. In blinding of outcome assessment, the study *Secreto et al. (2004)* was rated unclear owing to missing information, whereas those by *Nourozi et al. (2015)* and *Tranche et al. (2016)* were rated high risk owing to researchers' knowledge of assignments. In other bias evaluation, studies by *Costa et al. (2017)* and *Fontvieille, Dionne & Riesco (2017)* were rated high risk owing to the inclusion of non-pharmacological interventions (exercise); those by *Frigo et al. (2022)*, *Secreto et al. (2004)*, and *Nourozi et al. (2015)* were rated high risk owing to the inclusion of additional substances (*e.g.*, melatonin, calcium-D); and those by *Jou et al. (2008)* and *Tranche et al. (2016)* were rated high risk owing to potential placebo effects in open-label studies. The studies by *Atteritano et al. (2014)*, *Evans et al. (2011)*, *Imhof et al. (2018)*, and *Na Takuathung et al. (2024)* showed no relevant bias risks. The risk of bias in the included studies is presented in Figs. 2 and 3.

## Effect of soy isoflavones on menopausal symptoms

In all, seven studies measured menopausal symptoms, which represent the primary outcome of this systematic review. As shown in Table 1, the measurement of menopausal symptoms was performed using various scales, such as the Kupperman Index (KI) (*Tao et al., 2013*), Modified Kupperman Index (MKI) (*Tao et al., 2013*), Menopause Rating Scale (MRS) (*Dinger et al., 2006*), and Greene Climacteric Scale (GCS) (*Greene, 1998*). Random-effects meta-analysis (seven studies, 533 participants, with 267 in the soy isoflavones group and 266 in the control group) revealed that soy isoflavones have a certain therapeutic effect on menopausal symptoms (Hedges' g = $-0.25$, 95% CI [$-0.42$ to $-0.08$], $p = 0.00$), with a moderate effect size and low heterogeneity ($I^2 = 0.00\%$). After removing each study from the overall effect size, no significant differences were found. The forest plot is presented in Fig. 4. The bias analysis is shown in Fig. 5 and the sensitivity analysis is shown in Fig. 6.

Headache symptoms were measured through three studies, and a random-effects meta-analysis (three studies, 340 participants, 171 in the soy isoflavones group, and 169 in the control group) showed that soy isoflavones have some therapeutic effect on headaches, with a moderate effect size (Hedges' g = $-0.38$, 95% CI [$-0.60$ to $-0.17$], $p = 0.00$). The heterogeneity between studies was low ($I^2 = 0.00\%$). On removing each study and

**Table 1 Characteristics of all trials included in the present meta-analysis.**

| Author | Design | Country | N randomized subjects (Soy isoflavones, comparison) | Age (range) (Soy isoflavones, comparison) | Exposure | Daily dosage | Other compounds | Duration of active treatment | Outcome (measures) |
|---|---|---|---|---|---|---|---|---|---|
| *Atteritano et al. (2014)* | Double-blind RCT, parallel | Italy | 389 (198, 191 placebo) | (49–67) (53.00 ± 2.00, 52.00 ± 2.00) | Genistein | 54 mg | Calcium carbonate (500 mg) Vitamin D (400 IU) | 2 years | Zung Self-Rating Depression Scale Score[a] |
| *Jou et al., 2008* | Open label, parallel | China | 64 (34, 30 placebo) | 53.8 ± 3.8 EP group 53.6 ± 3.8 non-EP group 54.3 ± 2.8 placebo | Soy isoflavones | 135 mg | None | 6 months | Modified Kupperman Index[b] |
| *Nourozi et al. (2015)* | Double-blind RCT, parallel | Iran | 80 (40, 40 placebo) | (45–60) (52.13 ± 3.05, 51.39 ± 2.89) | Soy isoflavones | 185.55 mg | Calcium carbonate (500 mg) Vitamin D (200 IU) | 8 months | MENQOL[c] |
| *de Sousa-Munoz & Filizola (2009)* | Double-blind RCT, parallel | Brazil | 84 (42, 42 placebo) | (45–60) (53.35 ± 3.62) | Soy isoflavones | 120 mg | None | 16 weeks | CES-D[d] |
| *Costa et al. (2017)* | Double-blind RCT, parallel | Brazil | 36 (19, 17 placebo) | (45–60) (56.0 ± 1.3, 52.7 ± 1.3) | Soy isoflavones | 100 mg | None | 10 weeks | Kupperman Index[e] Menopause Rating Scale[f] Cervantes Scale[g] |
| *Frigo et al. (2022)* | Double-blind RCT, parallel | Brazil | 48 (24, 24 placebo) | (40–65) (51 ± 5.2, 50.7 ± 5.7) | Soy isoflavones | 40.5 mg | Phytoestrogens derived from flaxseed (40.9 mg) | 90 days | Kupperman Index |
| *Secreto et al. (2004)* | Double-blind RCT, parallel | Italy | 117 (58, 59 placebo) | (≥35) (52.0, 52.0) | Soy isoflavones | 80 mg | None | 3 months | Greene Climacteric Scale[h] Greene Psychological Subscale[i] |
| *Tranche et al. (2016)* | Open label, parallel | England | 90 (45, 45 placebo) | (≥45) (51.8 ± 3.1, 51.5 ± 3.5) | Soy isoflavones | 50 mg | None | 12 weeks | Menopause Rating Scale |
| *Evans et al. (2011)* | Double-blind RCT, parallel | Canada | 83 (41, 42 placebo) | (40–65) (53.39 ± 5.05, 53.50 ± 4.44) | Genistein | 30 mg | None | 12 weeks | Greene Climacteric Scale |
| *Imhof et al. (2018)* | Double-blind RCT, parallel | Austria Romania Germany | 192 (97, 95 placebo) | (40–70) (54.3 ± 6.4, 53.6 ± 5.3) | Soy isoflavones | 100 mg | None | 12 weeks | Greene Climacteric Scale |
| *Fontvieille, Dionne & Riesco (2017)* | Double-blind RCT, parallel | Canada | 31 (15, 16 placebo) | (50–70) (60.4 ± 3.4, 58.2 ± 5.7) | 44 mg of daidzein 16 mg of glycitein 10 mg of genistein | | None | 12 months | SF-36[j] Kupperman index PSS-10[k] |

**Table 1 (*continued*)**

| Author | Design | Country | N randomized subjects (Soy isoflavones, comparison) | Age (range) (Soy isoflavones, comparison) | Exposure | Daily dosage | Other compounds | Duration of active treatment | Outcome (measures) |
|---|---|---|---|---|---|---|---|---|---|
| *Na Takuathung et al. (2024)* | Double-blind RCT, parallel | Thailand | 100 (50, 50 placebo) | (48–65) (55.92 ± 3.52, 56.88 ± 3.62) | Soy isoflavones | 209 mg | Grape seed extract Tomato extract (Lycopersicon esculentum Mill) Vitamins Minerals | 12 weeks | Modified Kupperman Index MENQOL |

**Notes.**

[a] This is a self-reporting scale consisting of 20 items to assess the presence and severity of depressive symptoms.

[b] An adaptation of the original Kupperman Index, used to assess the severity of menopausal symptoms.

[c] The Menopause-Specific Quality of Life questionnaire is a tool designed to measure the impact of menopause on a woman's quality of life.

[d] The Center for Epidemiologic Studies Depression Scale is a self-reporting questionnaire used to measure symptoms of depression in the general population.

[e] A widely used scale to assess the severity of menopausal symptoms, which was one of the first tools developed for this purpose.

[f] A validated instrument used to evaluate the severity of climacteric symptoms and the effectiveness of treatments for menopausal symptoms.

[g] A 31- item questionnaire used to assess the health-related quality of life in women during the menopausal transition.

[h] A self-assessment tool consisting of 21 items across five dimensions to evaluate menopausal symptoms, including anxiety, depression, somatic symptoms, vasomotor symptoms, and sexual function.

[i] A part of the Greene Climacteric Scale that focuses on psychological symptoms such as anxiety and depression.

[j] A multipurpose, short-form health survey with 36 questions about perceptions of health and well-being.

[k] The Perceived Stress Scale-10 is a self-report measure developed to assess the global level of perceived stress.

analyzing the overall effect size, we found no significant difference in the results. The meta-analysis is presented in Fig. 7.

Paresthesia symptoms were measured through five studies, and a random-effects meta-analysis (five studies, 487 participants, with 246 in the soy isoflavone group and 241 in the control group) indicated that soy isoflavones had no significant effect in the treatment of Paresthesia symptoms, with a low effect size (Hedges' g = −0.16, 95% CI [−0.33 to 0.22], $p = 0.09$) and low heterogeneity among studies ($I^2 = 0.00\%$). On removing each study and analyzing the overall effect size, we did not find significant differences in the results. The meta-analysis is presented in Fig. 8.

Fatigue symptoms were measured across four studies, and the first data analysis showed high heterogeneity ($I^2 = 82.54\%$, $p = 0.08$). After evaluation, we considered that the heterogeneity originated from the article by *Imhof et al. (2018)* (owing to significant baseline differences), and therefore, we excluded this article from our analysis. The random-effects meta-analysis (including three studies, 212 participants: 108 in the soy isoflavone group and 104 in the control group) indicated that soy isoflavones were not significantly effective in treating fatigue symptoms, with a low effect size (Hedges' g = −0.17, 95% CI [−0.43 to 0.10], $p = 0.22$) and heterogeneity between studies ($I^2 = 0.00\%$). Meta-analysis is presented in Fig. 9.

The psychosocial symptoms were measured through five studies. A random-effects meta-analysis (five studies, 416 participants: 208 in the soy isoflavone group and 208 in the control group) indicated that soy isoflavones have a moderate effect on psychosocial symptoms, with a medium effect size (Hedges' g = −0.29, 95% CI [−0.48 to −0.10], $p = 0.00$) and low heterogeneity between studies ($I^2 = 0.00\%$). By removing each study and analyzing the overall effect size, we found no significant difference in the results. Meta-analysis is presented in Fig. 10.

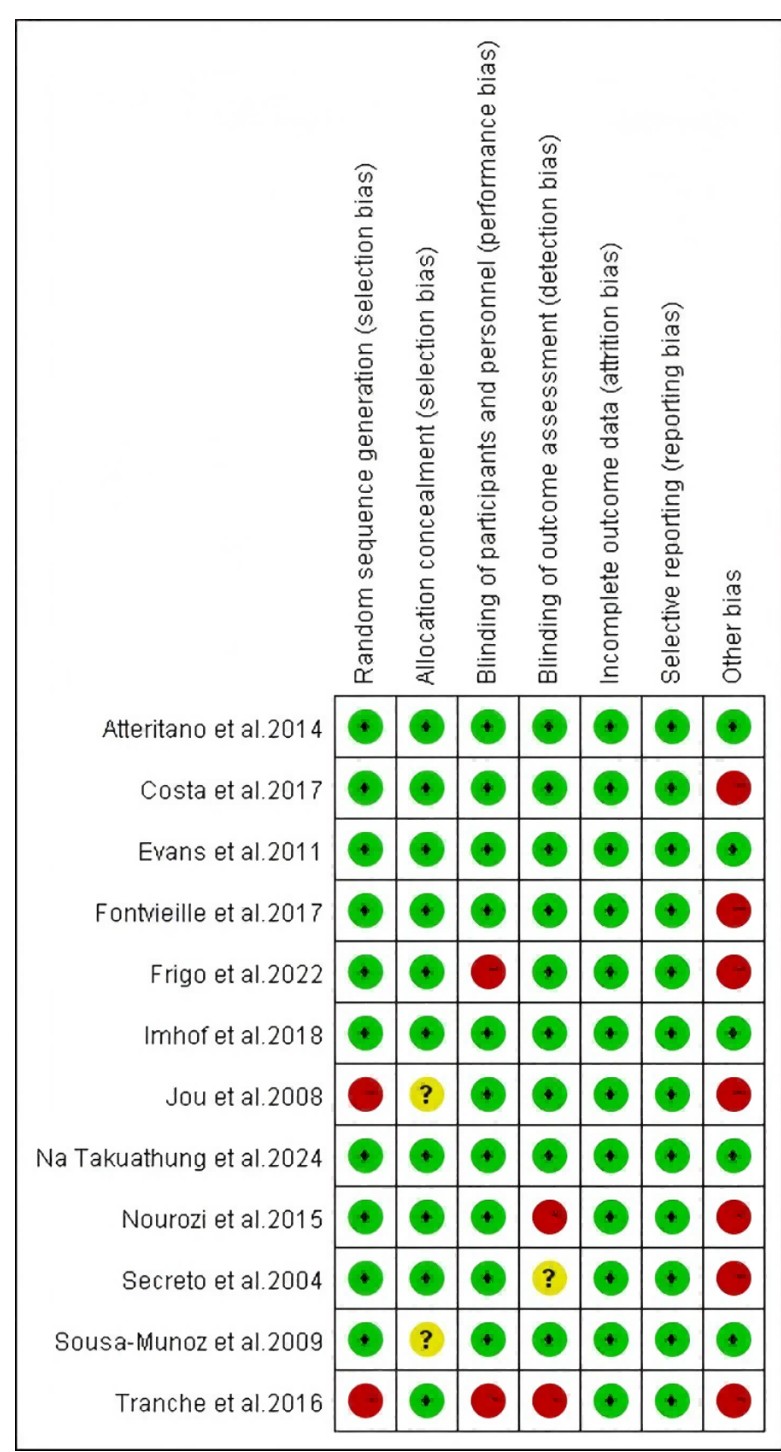

**Figure 2 Risk of bias assessment of the included studies.** Notes: *Atteritano et al., 2014*, *Costa et al., 2017*, *Evans et al., 2011*, *Fontvieille, Dionne & Riesco, 2017*, *Frigo et al., 2022*, *Imhof et al., 2018*, *Jou et al., 2008*, *Na Takuathung et al., 2024*, *Nourozi et al., 2015*, *Secreto et al., 2004*, *de Sousa-Munoz & Filizola, 2009*, *Tranche et al. (2016)*.

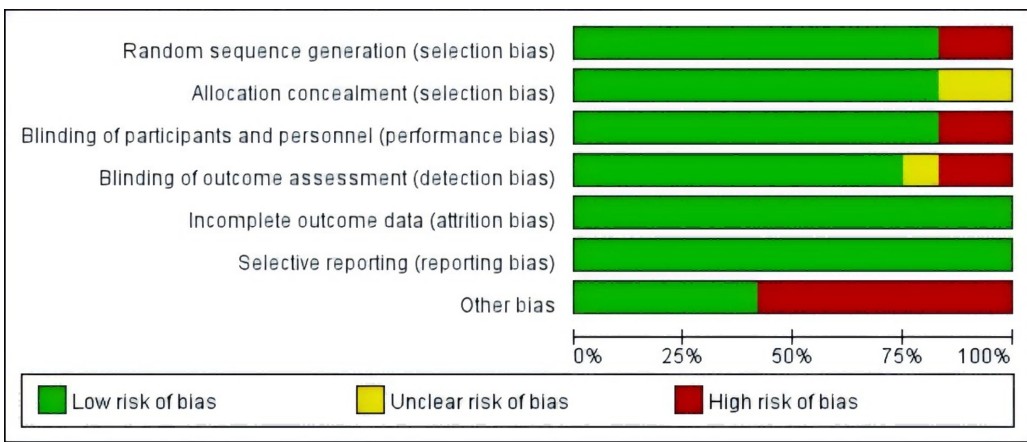

**Figure 3** Risk of bias assessment of the included studies.

| | | Treatment | | | Control | | | Hedges's g | Weight |
|---|---|---|---|---|---|---|---|---|---|
| Study | N | Mean | SD | N | Mean | SD | | with 95% CI | (%) |
| Jou et al.2008 | 34 | 4.32 | 4.65 | 30 | 6.74 | 6.35 | | -0.43 [ -0.92, 0.06] | 11.85 |
| Frigo et al.2022 | 24 | 9.8 | 4.6 | 24 | 11.1 | 6.7 | | -0.22 [ -0.78, 0.34] | 9.16 |
| Secreto et al.2004 | 58 | 8.9 | 8.4 | 59 | 9.2 | 9.9 | | -0.03 [ -0.39, 0.33] | 22.02 |
| Tranche et al.2016 | 45 | 11.8 | 7.3 | 45 | 14.4 | 7.1 | | -0.36 [ -0.77, 0.05] | 16.74 |
| Evans et al.2011 | 41 | 11.28 | 6.63 | 42 | 14.65 | 10.66 | | -0.38 [ -0.81, 0.06] | 15.43 |
| Fontvieille et al.2017 | 15 | 10.5 | 7.8 | 16 | 11.7 | 7 | | -0.16 [ -0.85, 0.53] | 6.05 |
| Na Takuathung et al.2024 | 50 | -0.76 | 3.54 | 50 | 0.1 | 4.45 | | -0.21 [ -0.60, 0.18] | 18.76 |
| **Overall** | | | | | | | | -0.25 [ -0.42, -0.08] | |

Heterogeneity: $\tau^2 = 0.00$, $I^2 = 0.00\%$, $H^2 = 1.00$

Test of $\theta_i = \theta_j$: Q(6) = 2.64, p = 0.85

Test of $\theta = 0$: z = -2.85, p = 0.00

Random-effects Hedges model

**Figure 4** Effect of soy isoflavones on menopausal symptoms. Notes: *Jou et al., 2008, Frigo et al., 2022, Secreto et al., 2004, Tranche et al. (2016), Evans et al., 2011, Fontvieille, Dionne & Riesco, 2017, Na Takuathung et al., 2024.*

Subsequently, physical symptoms were measured through three studies, and a random-effects meta-analysis (three studies, 211 participants: 105 in the soy isoflavone group and 106 in the control group) indicated that soy isoflavones have no therapeutic effect on physical symptoms, with a low effect size (Hedges' g $= -0.05$, 95% CI [$-0.37$ to 0.27], $p = 0.76$) and moderate heterogeneity between studies ($I^2 = 26.85\%$). By removing each study and analyzing the overall effect size, no significant difference was observed in the results. Meta-analysis is presented in Fig. 11.

Palpitation symptoms were measured through three studies, and a random-effects meta-analysis (three studies, 356 participants: 181 in the soy isoflavone group and 175 in the control group) indicated that soy isoflavones have a moderate effect on alleviating

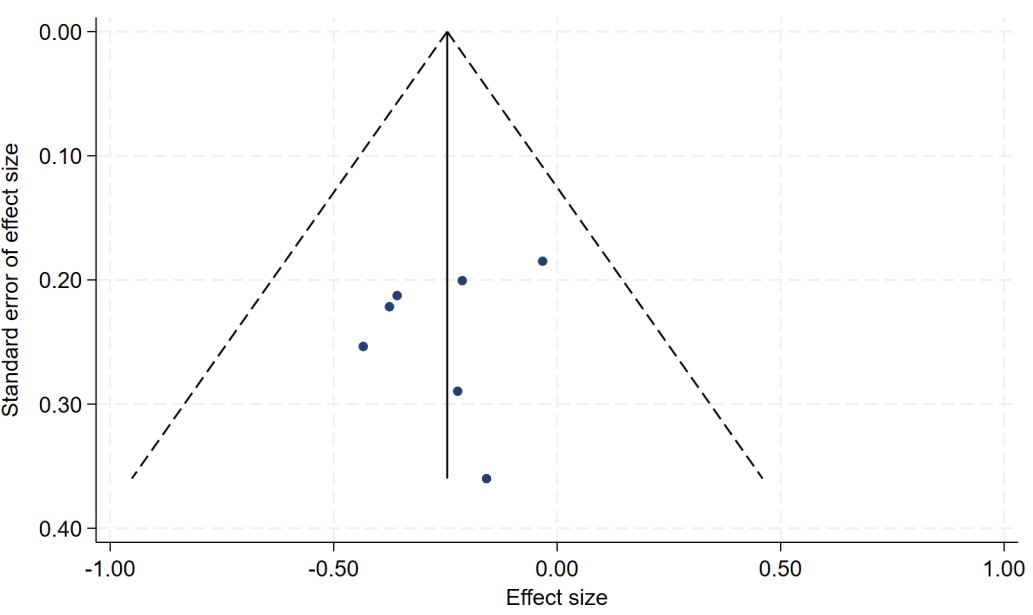

**Figure 5  Funnel plot of bias analysis.**

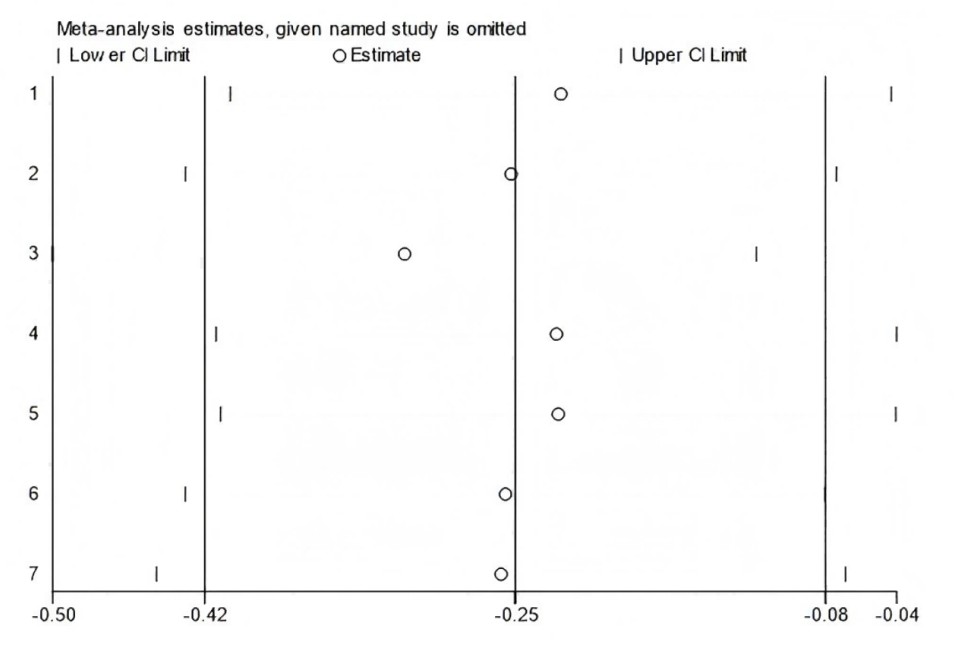

**Figure 6  Sensitivity analysis.**

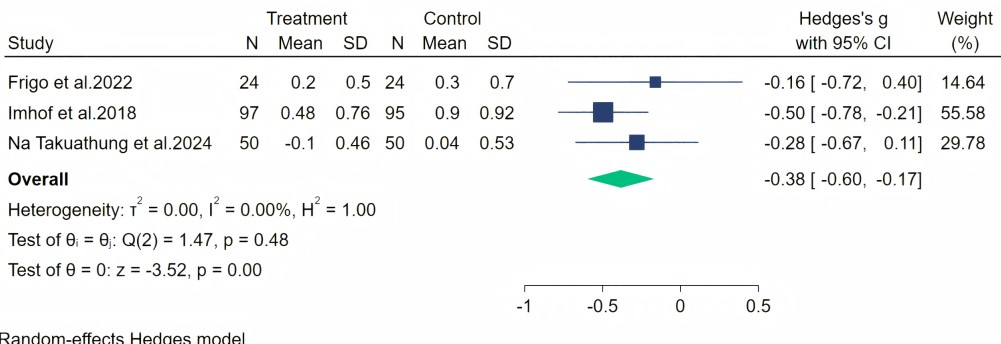

**Figure 7** **Effect of soy isoflavones on headache symptoms.** Notes: *Na Takuathung et al., 2024*, *Imhof et al. (2018)*, *Frigo et al., 2022*.

| Study | Treatment N | Mean | SD | Control N | Mean | SD | Hedges's g with 95% CI | Weight (%) |
|---|---|---|---|---|---|---|---|---|
| Jou et al.2008 | 34 | 0.19 | 0.6 | 30 | 0.37 | 0.79 | -0.26 [ -0.74, 0.23] | 13.16 |
| Frigo et al.2022 | 24 | 0.6 | 1.4 | 24 | 0.5 | 1.1 | 0.08 [ -0.48, 0.63] | 10.07 |
| Evans et al.2011 | 41 | 2.3 | 1.95 | 42 | 2.73 | 3 | -0.17 [ -0.60, 0.26] | 17.11 |
| Imhof et al.2018 | 97 | 0.36 | 0.72 | 95 | 0.54 | 0.86 | -0.23 [ -0.51, 0.06] | 39.05 |
| Na Takuathung et al.2024 | 50 | 0 | 0.4 | 50 | 0.04 | 0.86 | -0.06 [ -0.45, 0.33] | 20.62 |
| **Overall** | | | | | | | -0.16 [ -0.33, 0.02] | |

Heterogeneity: $\tau^2 = 0.00$, $I^2 = 0.00\%$, $H^2 = 1.00$
Test of $\theta_i = \theta_j$: Q(4) = 1.32, p = 0.86
Test of $\theta = 0$: z = -1.72, p = 0.09

Random-effects Hedges model

**Figure 8** **Effect of soy isoflavones on paresthesia symptoms.** Notes: *Jou et al., 2008*, *Frigo et al., 2022*, *Evans et al., 2011*, *Imhof et al., 2018*, *Na Takuathung et al., 2024*.

palpitation symptoms (Hedges' g $= -0.42$, 95% CI [$-0.63$ to $-0.22$], $p = 0.00$). The heterogeneity between studies was low ($I^2 = 0.00\%$). By removing each study and analyzing the overall effect size, we found no significant difference in the results. Meta-analysis is presented in Fig. 12.

Hot flashes were measured across four studies, and the initial analysis showed high heterogeneity ($I^2 = 86.67\%$, $p = 0.25$). Upon evaluation, we considered that the heterogeneity stemmed from the study by *Imhof et al. (2018)* (owing to significant baseline differences), and therefore, we excluded this article from our analysis. The random-effects meta-analysis (three studies, 143 participants: 73 in the soy isoflavone group and 70 in the control group) indicated that soy isoflavones are not effective in alleviating hot flashes, with a low effect size (Hedges' g $= -0.00$, 95% CI [$-0.33$ to 0.32], $p = 0.98$) and low heterogeneity between studies ($I^2 = 0.00\%$). Meta-analysis is presented in Fig. 13.

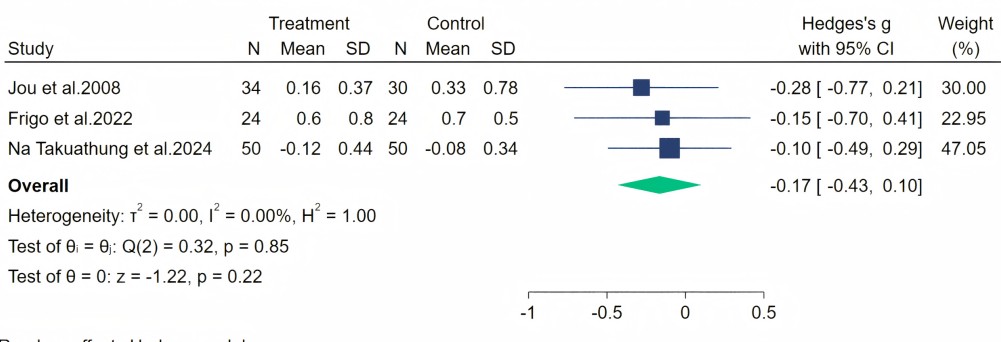

| Study | Treatment | | | Control | | | Hedges's g with 95% CI | Weight (%) |
|---|---|---|---|---|---|---|---|---|
| | N | Mean | SD | N | Mean | SD | | |
| Jou et al.2008 | 34 | 0.16 | 0.37 | 30 | 0.33 | 0.78 | -0.28 [ -0.77, 0.21] | 30.00 |
| Frigo et al.2022 | 24 | 0.6 | 0.8 | 24 | 0.7 | 0.5 | -0.15 [ -0.70, 0.41] | 22.95 |
| Na Takuathung et al.2024 | 50 | -0.12 | 0.44 | 50 | -0.08 | 0.34 | -0.10 [ -0.49, 0.29] | 47.05 |
| **Overall** | | | | | | | -0.17 [ -0.43, 0.10] | |

Heterogeneity: $\tau^2 = 0.00$, $I^2 = 0.00\%$, $H^2 = 1.00$

Test of $\theta_i = \theta_j$: Q(2) = 0.32, p = 0.85

Test of $\theta = 0$: z = -1.22, p = 0.22

Random-effects Hedges model

**Figure 9** **Effect of soy isoflavones on fatigue symptoms.** Notes: *Na Takuathung et al., 2024*, *Frigo et al., 2022*, *Jou et al., 2008*.

| Study | Treatment | | | Control | | | Hedges's g with 95% CI | Weight (%) |
|---|---|---|---|---|---|---|---|---|
| | N | Mean | SD | N | Mean | SD | | |
| Nourozi et al.2015 | 40 | 18.52 | 10.03 | 40 | 22.33 | 5.23 | -0.47 [ -0.91, -0.03] | 19.00 |
| Costa et al.2017 | 19 | 4 | 1.5 | 17 | 4.9 | 1.6 | -0.57 [ -1.22, 0.08] | 8.63 |
| Secreto et al.2004 | 58 | 4.4 | 5.9 | 59 | 4.8 | 7 | -0.06 [ -0.42, 0.30] | 28.38 |
| Evans et al.2011 | 41 | 5.48 | 3.91 | 42 | 7.65 | 6.68 | -0.39 [ -0.82, 0.04] | 19.86 |
| Na Takuathung et al.2024 | 50 | -0.003 | 0.94 | 50 | 0.25 | 1.11 | -0.24 [ -0.63, 0.15] | 24.14 |
| **Overall** | | | | | | | -0.29 [ -0.48, -0.10] | |

Heterogeneity: $\tau^2 = 0.00$, $I^2 = 0.00\%$, $H^2 = 1.00$

Test of $\theta_i = \theta_j$: Q(4) = 3.17, p = 0.53

Test of $\theta = 0$: z = -2.99, p = 0.00

Random-effects Hedges model

**Figure 10** **Effect of soy isoflavones on psychosocial symptoms.** Notes: *Nourozi et al., 2015*, *Costa et al., 2017*, *Secreto et al., 2004*, *Evans et al., 2011*, *Na Takuathung et al. (2024)*.

| Study | Treatment | | | Control | | | Hedges's g with 95% CI | Weight (%) |
|---|---|---|---|---|---|---|---|---|
| | N | Mean | SD | N | Mean | SD | | |
| Nourozi et al.2015 | 40 | 53.83 | 21.04 | 40 | 52.27 | 11 | 0.09 [ -0.34, 0.53] | 37.92 |
| Fontvieille et al.2017 | 15 | 93 | 8.8 | 16 | 89.7 | 13.4 | 0.28 [ -0.41, 0.97] | 18.54 |
| Na Takuathung et al.2024 | 50 | -0.42 | 0.92 | 50 | -0.09 | 1.14 | -0.32 [ -0.71, 0.08] | 43.53 |
| **Overall** | | | | | | | -0.05 [ -0.37, 0.27] | |

Heterogeneity: $\tau^2 = 0.02$, $I^2 = 26.85\%$, $H^2 = 1.37$

Test of $\theta_i = \theta_j$: Q(2) = 3.05, p = 0.22

Test of $\theta = 0$: z = -0.31, p = 0.76

Random-effects Hedges model

**Figure 11** **Effect of soy isoflavones on physical symptoms.** Notes: *Na Takuathung et al., 2024*, *Nourozi et al., 2015*, *Fontvieille, Dionne & Riesco, 2017*.

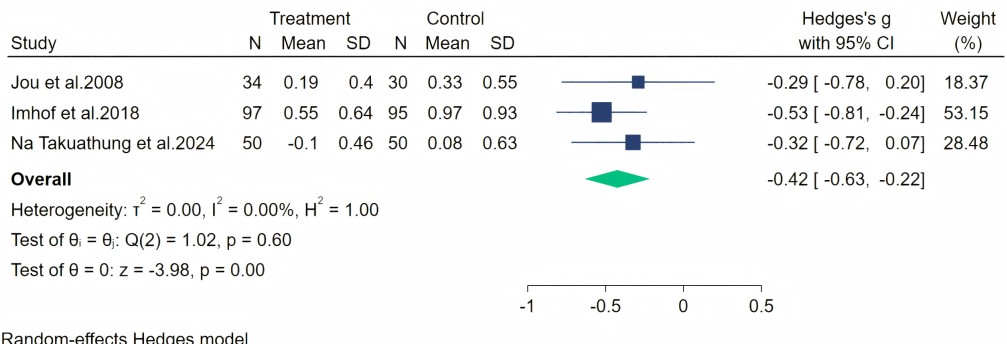

**Figure 12** **Effect of soy isoflavones on palpitation symptoms.** Notes: *Jou et al., 2008*, *Imhof et al., 2018*, *Na Takuathung et al., 2024*.

**Figure 13** **Effect of soy isoflavones on hot flashes symptoms.** Notes: *Jou et al., 2008*, *Imhof et al., 2018*, *Fontvieille, Dionne & Riesco, 2017*.

Furthermore, depression symptoms were measured through four studies. A random-effects meta-analysis (four studies, 748 participants: 378 in the soy isoflavone group and 370 in the control group) showed a significant therapeutic effect of soy isoflavones on depression symptoms, with a high effect size (Hedges' $g = -0.72$, 95% CI [$-1.17$ to $-0.28$], $p = 0.00$). However, the heterogeneity between studies was high ($I^2 = 86.76\%$). By removing the influence of each study on the overall effect size, no significant differences were found in the results. Meta-analysis is presented in Fig. 14. Considering the high heterogeneity and the limited number of included studies, we conducted meta-regression analyses separately for duration, dose, and population. The specific results are shown in Table 2.

Excessive sweating as a perimenopausal symptom was measured through two studies, and a random-effects meta-analysis (two studies, 95 participants: 49 in the soy isoflavone group and 46 in the control group) indicated that soy isoflavones were not significantly effective for excessive sweating symptoms, with a low effect size (Hedges' $g = -0.23$, 95%

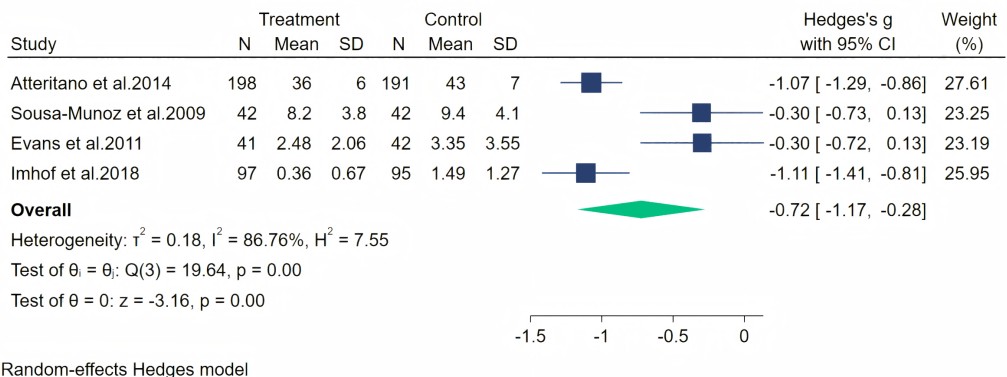

Figure 14 **Effect of soy isoflavones on depression symptoms.** Notes: *Atteritano et al., 2014*, *de Sousa-Munoz & Filizola, 2009*, *Evans et al., 2011*, *Imhof et al., 2018*.

Table 2 **Meta-regression of depression symptom.**

| Meta-regression of depression Variate | Estimate, 95% CI | *p*-value |
|---|---|---|
| Duration | −1.59 [−1.93 to 0.89] | 0.254 |
| Dose | −1.03 [−3.53 to 2.16] | 0.411 |
| Population | −0.77 [−1.63 to 1.14] | 0.523 |
| Baseline symptom severity | −1.50 [−2.33 to 1.12] | 0.271 |

Figure 15 **Effect of soy isoflavones on excessive sweating symptoms.**

CI [−0.62 to 0.17], $p = 0.27$) and low heterogeneity between studies ($I^2 = 0.00\%$). By removing the influence of each study on the overall effect size, no significant differences were found in the results. Meta-analysis is presented in Fig. 15.

Insomnia symptoms were measured through two studies, and a random-effects meta-analysis (two studies, 148 participants: 74 in the soy isoflavone group and 74 in the control group) indicated that soy isoflavones have no significant effect on insomnia, with a low

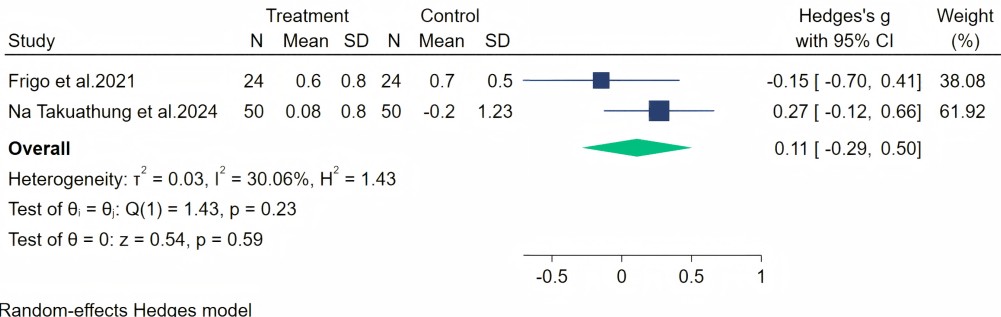

**Figure 16** **Effect of soy isoflavones on insomnia symptoms.** Notes: *Frigo et al., 2022*, *Na Takuathung et al., 2024*.

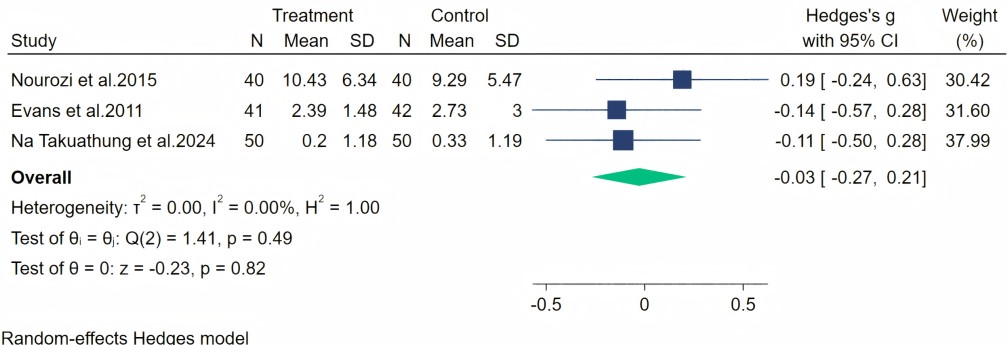

**Figure 17** **Effect of soy isoflavones on vasomotor.** Notes: *Nourozi et al., 2015*, *Evans et al., 2011*, *Na Takuathung et al., 2024*.

effect size (Hedges' $g = 0.11$, 95% CI [$-0.29$ to $0.50$], $p = 0.59$) and moderate heterogeneity between studies ($I^2 = 30.06\%$). By removing the influence of each study on the overall effect size, no significant differences were found in the results. Meta-analysis is presented in Fig. 16.

Vasomotor symptoms were measured through three studies, and a random-effects meta-analysis (three studies, 263 participants: 131 in the soy isoflavone group and 132 in the control group) indicated that soy isoflavones are not significantly effective in treating vasomotor symptoms, with a low effect size (Hedges' g $= -0.03$, 95% CI [$-0.27$ to $0.21$], $p = 0.82$). Heterogeneity between studies was low ($I^2 = 0.00\%$). By removing the influence of each study on the overall effect size, we did not find any significant differences in the results. Meta-analysis is presented in Fig. 17.

## DISCUSSION

Soy isoflavones, including genistein, daidzein, and glycitein, are recognized for their multifaceted pharmacological actions, particularly with regard to anti-inflammation,

antioxidant activity, and menopausal symptom alleviation. These compounds exhibit anti-inflammatory effects by modulating key signaling pathways and reducing the release of inflammatory mediators, which are crucial for managing inflammation-related menopausal symptoms. Their antioxidant properties are highlighted by their ability to scavenge reactive oxygen species (ROS), thereby protecting against oxidative damage and the aging process. Furthermore, soy isoflavones have been shown to alleviate menopausal symptoms by acting as phytoestrogens, binding to estrogen receptors and exerting both agonistic and antagonistic effects, which can help reduce and alter vasomotor symptoms. Recent research has also underscored their potential in preventing bone loss (*Akhlaghi et al., 2020*), reducing cancer risk (*Fan et al., 2022*), and exhibiting anti-obesity and anti-diabetes effects (*Dwivedi et al., 2024*). These findings position soy isoflavones as key bioactive compounds in the management of menopause and promotion of women's health.

Our research findings indicate that soy isoflavones have a certain positive effect on menopausal symptoms, which are very commonly observed in perimenopausal women. In particular, the Hedges' g value, indicating the overall effect size for menopausal symptoms, was −0.25 (95% CI [−0.42 to −0.08], $p = 0.00$), indicating a small but statistically significant reduction in symptom severity. However, this effect size is considered small according to Cohen's criteria (*Gignac & Szodorai, 2016*), and its clinical significance remains uncertain. For example, although a reduction in symptoms was observed, the magnitude of this reduction may not be sufficient to result in substantial improvements in quality of life for all individuals. Similarly, the effect size was also small for psychosocial symptoms (Hedges' g = −0.29), suggesting that while soy isoflavones may provide some benefit, the impact may be limited in a clinical context. The effect size was larger for depression symptoms (Hedges' g = −0.72), indicating a potentially more meaningful clinical benefit, although this finding was accompanied by high heterogeneity among studies ($I^2 = 86.76\%$). This indicates that although soy isoflavones may be effective for some individuals, the variability in response may limit their broad applicability.

Soy isoflavones appear to modulate the NF-κB signaling pathway, a critical mediator in inflammatory and immune responses (*Ma et al., 2021*). This pathway modulation can be beneficial in reducing chronic inflammation, thereby potentially reducing the risks associated with inflammatory diseases. Furthermore, the interaction of soy isoflavones with the JAK-STAT pathway, important in cellular processes such as growth and apoptosis, suggests their role in cancer management (*Wu et al., 2021*). Soy isoflavones can help inhibit undesirable cell proliferation and metastasis by influencing this pathway, thus offering a complementary approach to traditional cancer therapies. Skin health may also benefit from the inclusion of soy isoflavones in the diet or skincare regimen. Their ability to counteract UV-induced damage by promoting DNA repair and reducing the expression of inflammatory genes such as GADD45 and COX-2 is significant (*Ou et al., 2021*). Such protective and reparative actions not only help in preventing skin cancer but also combat premature aging. Enhanced collagen synthesis and reduced inflammatory cytokines contribute to improved skin elasticity and reduced wrinkle formation, manifesting as visible anti-aging effects. Moreover, PI3K-Akt signaling pathway regulation by soy isoflavones highlights their role in cell survival and proliferation (*Khezri et al., 2022*). The

involvement of this pathway in cancer cell growth and survival underscores the potential of soy isoflavones in influencing tumor dynamics, possibly offering a natural adjunct therapy for cancer management. Potential interaction with drug metabolism, particularly through cytochrome P450 enzyme modulation, indicates that the clinical use of soy isoflavones deserves critical consideration (*Ronis, 2016*). As these enzymes play a significant role in the metabolism of many drugs, soy isoflavones can alter the pharmacokinetics of concurrent medications, necessitating careful management and monitoring by healthcare providers.

Our research findings indicate that soy isoflavones have a positive effect on menopausal symptoms, which are very common in perimenopausal women. This finding contradicts a recent study (*Gencturk, Bilgic & Kaban, 2024*), reporting that soy isoflavones have no effect on menopausal symptoms (including vasomotor, psychosocial, physical, sexual, and urogenital discomfort) or quality of life in postmenopausal women. However, we found that soy isoflavones have a significant therapeutic effect on depression symptoms. We consider that the discrepancy may arise from differences in search term settings and inclusion criteria, which might have led to a relatively small number of studies being included and potentially biased the results.

This study was limited to only 12 articles, with the evaluation of menopausal symptoms conducted in only seven trials. To better understand the potential benefits of soy isoflavones as a complementary treatment for menopausal symptoms, further research is necessary. Another significant limitation concerns the patient population; all included studies typically featured small sample sizes. Given the accessibility and favorable tolerance of soy isoflavones among patients, future studies should aim to include larger cohorts. Moreover, the inability to access databases for all trials hindered our capacity to conduct an individual patient data analysis, which would have provided a more thorough evaluation. It is important to highlight that most studies were conducted in European countries, where soy consumption is relatively low. Investigating the effects of soy isoflavones in Asian countries, where soy foods are more commonly consumed, would be an intriguing direction for future research. It is also crucial to point out that the primary reason soy isoflavones are generally not recommended for breast cancer patients is their potential for bidirectional hormonal modulation and their effects on estrogen receptors. Given their structural resemblance to estrogen, soy isoflavones can bind to estrogen receptors and produce similar effects. In breast cancer patients, particularly those with estrogen receptor-positive (ER+) tumors, soy isoflavones may imitate estrogen's actions and promote the proliferation of breast cancer cells (*Yamashita et al., 2022*). Moreover, certain studies suggest that at lower concentrations, soy isoflavones may not inhibit but actually stimulate the growth of estrogen-dependent breast cancer cells, such as MCF-7 (*Uifalean et al., 2015*). Other research has indicated a dose-dependent relationship, wherein increased levels of soy isoflavones from soy protein are associated with enhanced growth of estrogen-dependent tumor cells (*Johnson et al., 2016*). Continued investigation is vital to clarify these contentious findings.

Adverse effects and contraindications of soy isoflavones should be carefully considered in clinical practice. With regard to adverse effects, some individuals may experience gastrointestinal discomfort such as nausea, abdominal pain, or diarrhea after consuming soy isoflavones (*Al-Nakkash & Kubinski, 2020*). Allergic reactions, including skin itching,

redness, and rashes, may also occur among individuals allergic to soy products (*Wang, He & Raghavan, 2023*). In addition, high doses of soy isoflavones potentially affect liver, kidney, and reproductive system functions in animal models (*Neshatbini Tehrani et al., 2024*). However, these findings have not been conclusively translated to humans. Regarding contraindications, soy isoflavones should be avoided by individuals with a history of estrogen-related diseases, such as breast hyperplasia, breast cancer, and uterine cancer, as they may exacerbate these conditions. The currently recommended intake of soy isoflavones for perimenopausal women is mostly in the range of 50–100 mg per day (*Kang et al., 2022*; *Khapre, Deshmukh & Jain, 2022*). Prepubescent girls, pregnant and breastfeeding women, and individuals with hypothyroidism should also refrain from using soy isoflavones owing to the potential adverse effects on growth, development, and thyroid function (*Huser et al., 2018*). Furthermore, patients administering anti-estrogen medications should avoid soy isoflavones to prevent interference with their treatment. In clinical practice, it is essential to weigh the potential benefits of soy isoflavones against the risks and to closely monitor patients for any adverse reactions.

In conclusion, our study suggests that soy isoflavones may have some potential benefits in relieving certain menopausal symptoms, particularly depressive symptoms. However, the small effect sizes for many outcomes, wide confidence intervals, and methodological inconsistencies among the included studies limit the certainty of these findings. The clinical significance of these results remains uncertain, and further research is needed to better understand the potential benefits and risks of soy isoflavones for the management of menopausal symptoms. Future studies should focus on larger sample sizes, standardized intervention protocols, and the inclusion of diverse populations to provide more robust evidence. In addition, investigating the potential adverse effects and contraindications of soy isoflavones is important to inform clinical practice and ensure patient safety.

### Funding

This work was supported by the Innovation and Entrepreneurship Project of Inner Mongolia Medical University of China (No. S202410132019), the Inner Mongolia Medical University Innovation Team Project (No. YKD2022TD031), the Inner Mongolia Medical University Youth Project (No. 2022QN005) and the Natural Science Key Project of Higher Education Research in Inner Mongolia Autonomous Region (No. NJZZ22686). The funders had no role in study design, data collection and analysis, decision to publish, or preparation of the manuscript.

### Grant Disclosures

The following grant information was disclosed by the authors:
The Innovation and Entrepreneurship Project of Inner Mongolia Medical University of China: No. S202410132019.
The Inner Mongolia Medical University Innovation Team Project: No. YKD2022TD031.
The Inner Mongolia Medical University Youth Project: No. 2022QN005.

The Natural Science Key Project of Higher Education Research in Inner Mongolia Autonomous Region: No. NJZZ22686.

## Competing Interests

The authors declare there are no competing interests.

## Author Contributions

- Haodi Luan conceived and designed the experiments, performed the experiments, analyzed the data, prepared figures and/or tables, authored or reviewed drafts of the article, and approved the final draft.
- Qianqian Liu performed the experiments, analyzed the data, prepared figures and/or tables, authored or reviewed drafts of the article, and approved the final draft.
- Yahui Guo performed the experiments, analyzed the data, prepared figures and/or tables, authored or reviewed drafts of the article, and approved the final draft.
- Hua Fan performed the experiments, analyzed the data, prepared figures and/or tables, authored or reviewed drafts of the article, and approved the final draft.
- Sileng A. performed the experiments, analyzed the data, prepared figures and/or tables, authored or reviewed drafts of the article, and approved the final draft.
- Jing Lin conceived and designed the experiments, performed the experiments, analyzed the data, prepared figures and/or tables, authored or reviewed drafts of the article, and approved the final draft.

## Data Availability

This is a systematic review/meta-analysis.

## Supplemental Information

Supplemental information for this article can be found online at http://dx.doi.org/10.7717/peerj.19715#supplemental-information.

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
