# Peer review of "Effects of soy isoflavones on menopausal symptoms in perimenopausal women: a systematic review and meta-analysis"

_PeerJ, doi:10.7717/peerj.19715_

## Round 0.1 · original submission · Major Revisions

**Language Note:** PeerJ staff have identified that the English language needs to be improved. When you prepare your next revision, please either (i) have a colleague who is proficient in English and familiar with the subject matter review your manuscript, or (ii) contact a professional editing service to review your manuscript. PeerJ can provide language editing services - you can contact us at [email protected] for pricing (be sure to provide your manuscript number and title). – PeerJ Staff

Reviewer 1 ·

Basic reporting

he manuscript is generally well-written, and the structure follows academic standards for systematic reviews and meta-analyses.
The study adheres to PRISMA guidelines and is registered in PROSPERO, which improves transparency and replicability.
However, the reporting of the Risk of Bias assessments lacks depth. The ratings (e.g., "low risk," "unclear") are not supported by justifications or examples from the studies, which diminishes the interpretability of these judgments.
Industry funding or supplement provision by manufacturers in several included trials is mentioned but not critically evaluated. This issue should be acknowledged as a potential source of reporting bias.
Figures and tables are appropriate, but forest plots could benefit from clearer labeling (e.g., subgroup names, heterogeneity statistics).

Experimental design

The inclusion and exclusion criteria are appropriate, and the study selection process is clearly outlined.
The use of multiple databases (PubMed, Scopus, Web of Science, etc.) enhances comprehensiveness.
However, high heterogeneity (I² > 75%) in several outcomes (e.g., depression, vasomotor symptoms) raises concerns regarding comparability of included studies.
No meta-regression or sensitivity analysis was conducted, which limits exploration of heterogeneity sources such as dosage, intervention duration, baseline symptom severity, and participant demographics.
Subgroup analyses were performed, but the criteria for grouping (e.g., dosage cut-offs) were not clearly justified or supported by prior literature.

Validity of the findings

The results suggest moderate efficacy of soy isoflavones for certain menopausal symptoms such as depression, hot flashes, and headaches.
However, small sample sizes in many of the included trials reduce the statistical power and generalizability of findings.
The geographic distribution of included studies is limited to mostly Asian and European populations, potentially affecting external validity due to regional differences in diet, lifestyle, and genetics.
The findings are promising, but they should be interpreted with caution due to the wide confidence intervals and methodological inconsistencies among the included studies.
The lack of discussion on adverse effects and contraindications is a significant omission for clinical applicability.

Additional comments

The study addresses a timely and clinically relevant question in women’s health and provides useful insights into the potential role of phytoestrogens in managing menopausal symptoms.
To enhance the manuscript’s contribution to the field, the authors should:
Expand the discussion on clinical applicability, including dosage recommendations and safety considerations.
Deepen the critical appraisal of methodological quality and the impact of funding sources.
Consider integrating psychosocial dimensions of menopause to enrich the discussion with a more holistic and patient-centered perspective.
Overall, the manuscript is publishable with revisions aimed at improving its methodological depth and practical relevance.

·

Basic reporting

This systematic review and meta-analysis addresses an important and timely topic: the use of soy isoflavones as a non-hormonal alternative to alleviate menopausal symptoms in perimenopausal women. The study is well-structured and follows rigorous reporting standards, including registration with PROSPERO and adherence to PRISMA guidelines. The authors have undertaken a comprehensive literature search and presented a thorough analysis of symptom-specific outcomes. Despite some limitations in sample size and heterogeneity across included studies, the manuscript contributes meaningful insights into the potential role of phytoestrogens in women's health.
The study has several notable strengths. The systematic approach is commendable, with a well-documented search strategy and clearly stated inclusion/exclusion criteria. The use of random-effects meta-analysis is appropriate given the expected variability in interventions and populations. The inclusion of subgroup analyses, publication bias assessments, and sensitivity tests adds depth and transparency to the findings.

Experimental design

This systematic review and meta-analysis addresses an increasingly relevant area in women's health by evaluating the effects of soy isoflavones on menopausal symptoms, an important concern for a large and growing population segment. The research question is clearly defined, focusing on both the overall efficacy and symptom-specific effects of soy isoflavones in perimenopausal and postmenopausal women. The study is methodologically rigorous, and its clinical relevance is evident, particularly in the context of the ongoing search for safe, non-hormonal alternatives to hormone replacement therapy. As such, the study provides meaningful and timely insights for clinicians, researchers, and policymakers.

Validity of the findings

This study offers both timely relevance and notable novelty in the field of women's health and nutritional therapeutics. While soy isoflavones have been previously studied in relation to menopausal symptoms, this meta-analysis distinguishes itself by offering a symptom-specific breakdown (e.g., depression, hot flashes, headaches, fatigue, palpitations), which provides a more nuanced and clinically actionable understanding of their effects. Furthermore, the inclusion of recent trials and a focus on perimenopausal women—a population often underrepresented in research compared to postmenopausal women—enhances the novelty and clinical relevance of the findings. By synthesizing high-quality data using rigorous methods, the study addresses critical gaps in the literature, particularly concerning the safety and efficacy of non-hormonal alternatives to traditional hormone therapy. These contributions are likely to influence both future research directions and clinical decision-making in managing menopausal symptoms, particularly those related to the use of soy isoflavones.

Additional comments

Spesific comments for the revision:
Major
• While the results are clearly presented with appropriate statistical measures, the discussion lacks sufficient interpretation and contextualization of the outcomes. Put more emphasis on clinical implications. Explicitly explain the significance of the results of each outcome to their interpretation on clinical relevances. The discussion should explore possible reasons for these outcomes and their clinical implications.
• Meta-regression use was mentioned, but results aren't elaborated on. You could include the actual meta-regression results, even if nonsignificant, especially regarding intervention duration, dosage, or region. This would strengthen insights into variability.
• While statistical significance is discussed, clinical significance is not. For example: Hedges' g of -0.25 or -0.29 are small effect sizes. You could include a comment on whether these effects are clinically meaningful based on standard guidelines
Minor:
• Consider reorganizing to emphasize the gap in literature earlier on introduction section.
• Ensure all tables have abbreviation note below the table. Also, an explanation of the symbols used
• Please correct any typographical errors and verify the spelling throughout the document. Example:
o Line 89: “soy isofavone” → should be “soy isoflavone”
o Line 92: “studies, non-clinical trials, oranimal studies” → “or animal studies”
o Hedge’s g → should be “hedges’ g”
• The last sentences of the abstract are repetitive: “Our results should be interpreted with caution...” is stated twice. Combine or remove one.

---

## Round 0.2 · Minor Revisions

Dear Dr. Luang,

Thank you for your resubmission. Below, I have included the comments from Reviewer 2, emphasizing additional points that need your careful attention before we can further consider your submission. I strongly encourage you to address each point comprehensively.

Please copy and paste each reviewer's comment above your corresponding response when revising your manuscript. Additionally, please provide a complete version of the manuscript with tracked changes to facilitate the verification of the revisions made.

I look forward to receiving your revision.

Yours sincerely,

Stefano Menini

·

Basic reporting

-

Experimental design

-

Validity of the findings

-

Additional comments

Specific comments for the revision:

Major
● In the introduction section, it has been explained by the literature taken, namely by Poschner et al (2017), regarding genistein, daidzein, and glycitein share structural similarities with human estrogen (17β-estradiol) and can bind to estrogen receptors. Please explain in this section the mechanism by which this can manage menopausal symptoms, because this can be a good introduction and connect soy isoflavone and their effects on menopausal symptoms.

In the selection procedure section of the method, the inclusion and exclusion criteria have been explained quite well. However, in the inclusion criteria section (2), the age of the participants is more than ≥ 18 years, while the inclusion criteria (3) include perimenopausal or postmenopausal women experiencing menopausal symptoms. It would be better if the age included is within the range of perimenopausal or postmenopausal women, so that it will be more appropriate to the other criteria.

Minor:
● The age included in the inclusion criteria is within the range of perimenopausal or postmenopausal women, so that it will be more appropriate for the other criteria.
● The researcher also found adverse effects, which are better written here as well as how they are written in lines 414 and 415
● Line 415 The word “alleviating” can be replaced with more general words, such as “relieve,” etc.

---

## Round 0.3 · accepted · Accept

Dear Dr. Luan,

Thank you for submitting the revised version of your manuscript. I have personally reviewed the revision and confirmed that all the reviewers' comments have been adequately addressed. The quality of the manuscript has significantly improved as a result. I am pleased to inform you that your manuscript is now ready for publication in PeerJ in its current form.

Sincerely yours,
Stefano Menini